# Epigenetics in Heart Failure

**DOI:** 10.3390/ijms252212010

**Published:** 2024-11-08

**Authors:** Jamie Sin Ying Ho, Eric Jou, Pek-Lan Khong, Roger S. Y. Foo, Ching-Hui Sia

**Affiliations:** 1Department of Cardiology, National University Heart Centre, Singapore 119228, Singapore; sinyingh@gmail.com (J.S.Y.H.); roger.foo@nus.edu.sg (R.S.Y.F.); 2Department of Oncology, University of Oxford, Oxford OX3 7DQ, UK; eric.jou@oncology.ox.ac.uk; 3Kellogg College, University of Oxford, Oxford OX2 6PN, UK; 4Department of Diagnostic Imaging, National University Hospital, National University Health System, Singapore 119074, Singapore; dnrkpl@nus.edu.sg; 5Department of Diagnostic Radiology, Yong Loo Lin School of Medicine, National University of Singapore, Singapore 119074, Singapore; 6Cardiovascular Research Institute, Yong Loo Lin School of Medicine, National University of Singapore, Singapore 117597, Singapore

**Keywords:** heart failure, epigenetic modification, personalized medicine, histone modifications, DNA methylation

## Abstract

Heart failure is a clinical syndrome with rising global incidence and poor prognosis despite improvements in medical therapy. There is increasing research interest in epigenetic therapies for heart failure. Pathological cardiac remodelling may be driven by stress-activated cardiac signalling cascades, and emerging research has shown the involvement of epigenetic signals that regulate transcriptional changes leading to heart failure. In this review, we appraise the current evidence for the role of epigenetic modifications in heart failure. These include DNA methylation and histone modifications by methylation, acetylation, phosphorylation, ubiquitination and sumoylation, which are critical processes that establish an epigenetic pattern and translate environmental stress into genetic expression, leading to cardiac remodeling. We summarize the potential epigenetic therapies currently in development, including the limited clinical trials of epigenetic therapies in heart failure. The dynamic changes in the epigenome in the disease process require further elucidation, and so does the impact of this process on the development of therapeutics. Understanding the role of epigenetics in heart failure may pave the way for the identification of novel biomarkers and molecular targets, and facilitate the development of personalized therapies for this important condition.

## 1. Introduction

Heart failure is a clinical syndrome with symptoms or signs caused by a structural or functional cardiac abnormality, corroborated by elevated natriuretic peptide levels or objective evidence of pulmonary or systemic congestion. The prevalence of heart failure has been increasing globally, affecting around 64 million people worldwide with an overall prevalence of 1–3% [1]. Despite advancements in medical treatment and healthcare access, the mortality rate following heart failure hospitalization is around 20% at 1 year and 50% at 5 years [2]. Therefore, increasing research has focused on the mechanisms underlying the development of heart failure to identify additional strategies to treat this progressive disease.

Epigenetic regulation of gene expression impacts the phenotype without altering the genotype [3] and may influence gene activity at the transcriptional, post-transcriptional, translational and post-translational levels. These modifications include DNA methylation, histone modifications such as methylation, acetylation and phosphorylation, and chromatin remodelling. Epigenetics allows non-genetic cellular memory to accumulate over the organism’s lifetime, which may be expressed in specific developmental and environmental situations and has been shown to be involved in diseases such as cancer, inherited diseases and cardiovascular diseases [4]. Epigenetic changes often occur early in the disease pathology and have the potential to be used as a biomarker for disease diagnosis, progression and prognosis. In the heart, cardiomyocytes, fibroblasts, endothelial cells and progenitor cells are under tight epigenetic regulation, which mediates a dynamic adaptation to environmental challenges and response to biochemical stress. In heart failure, pathological cardiac remodelling may be driven by stress-activated cardiac signalling cascades, and emerging research has shown an involvement of epigenetic signals that regulate transcriptional changes that lead to heart failure.

In this review, we appraise the current evidence for the role of epigenetic modifications in heart failure. Understanding the role of epigenetics in heart failure may pave the way for the identification of novel molecular targets and facilitate the development of personalized therapies.

## 2. Pathophysiology of Heart Failure

Heart failure is a clinical syndrome of typical signs and symptoms caused by structural and/or functional cardiac abnormality, leading to a reduced cardiac output and/or elevated intracardiac pressure at rest or during stress. Heart failure is initiated by underlying cardiac dysfunction, for example myocardial infarction (MI), valve abnormalities, arrhythmias, endocardial and pericardial disease, and may present with reduced left ventricular ejection fraction (HFrEF; EF < 40%), mid-range ejection fraction (HFmrEF; EF 40–50%) or preserved ejection fraction (HFpEF; EF > 50%). In HFpEF, structural remodelling such as cardiomyocyte hypertrophy and intracellular fibrosis causes the inability of the left ventricle to relax. In HFrEF, cardiomyocyte loss due to myocardial infarction or overload leads to the impairment of ventricular contraction. Overload, which causes an increase in pressure and volume in the heart, results in structural and functional adaptations, leading to a vicious cycle of processes in the chronic setting with further deterioration in heart function. This includes the activation of the sympathoadrenergic system and renin–angiotensin–aldosterone system and the release of vasoactive peptides, all of which may further worsen cardiac function. This is due to maladaptive mechanisms in response to chronic stimulation, such as the desensitization of the β-adrenergic system, alerting of the intracellular calcium handling system, and shifting of the Frank–Starling curve to a negative force–frequency relationship. The re-expression of fetal gene program associated with myocyte hypertrophy and heart failure may be induced by cardiac stressors in the adult heart, resulting in epigenetic changes early in the disease process which underpin the structural and functional changes that result in heart failure.

## 3. DNA Methylation in Heart Failure

Research focused on the role of DNA methylation in the pathophysiology of heart failure has received increased attention in recent years (Figure 1 and Figure 2, Table 1). There are various types of genomic DNA methylation, and the most commonly known is the addition of methyl group to the 5′ position of cytosine (5mC) by de novo methyltransferases (DNMTs), which occurs mainly in the cytosinephosphateguanine (CpG) context [5]. Classically, DNA methylation promotes gene silencing by blocking transcription factors from binding to the DNA directly or by allowing the binding of methyl-binding proteins (MBPs), which cause gene repression through interactions with a co-repressor complex. Conversely, the oxidation of 5mC by the ten-eleven translocation (TET) family of proteins generates 5-hydroxymethylcytosine (5hmC), which is associated with transcription activation. DNA methylation enzymes are broadly divided into writing, erasing and reading enzymes. Examples of writing enzymes include DNMT1, which adds methyl groups to cytosine residues mimicking the original DNA methylation pattern before DNA replication with high fidelity, while DNMT3a and DNMT3b are de novo methyltransferases that add methyl groups to naked DNA and establish new methylation patterns. Eraser enzymes remove methyl groups, while reading enzymes such as MBPs, UHRF (ubiquitin-like, containing PHD and RING finger domain) proteins and zinc finger domain proteins recognize methyl groups and affect transcription factor binding, which can lead to gene silencing or activation [5].

Targeted DNA methylation profiling of human cardiac tissues in patients with heart failure from hypertrophic obstructive cardiomyopathy, ischemic cardiomyopathy, dilated cardiomyopathy and control patients showed variable DNA methylation. A recent study revealed 195 uniquely differentially methylated regions [27], and the gene and non-coding RNA expression analysis found four hypermethylated genes (HEY2, MSR1, MYOM3 and COX17) and two hypomethylated genes (connective tissue growth factor [CTGF] and matrix metalloprotease 2 [MMP2]), with the corresponding upregulation or downregulation of genetic expression. The hypomethylation of MMP2 and CTGF in dilated cardiomyopathy increased the expression of MMP2, a regulator of collagen turnover and fibrosis that is increased in patients with heart failure, and CTGF, which functions in extracellular matrix fibrosis and is highly expressed in failing hearts [6]. The main de novo DNA methyltransferase in cardiomyocytes is DNMT3A, which plays an important role in cardiac stress, with effects on glucose and lipid metabolism and cardiac contraction. DNMT3A knocked out in human cardiomyocytes resulted in impaired contraction kinetics associated with a higher atrial gene expression and lower MYH7/MYH6 ratio [28]. MYH7 encodes β-myosin heavy chain (MHC), which is the major left ventricular MHC in the adult human, while MYH6 encodes α-MHC and is the major MHC in the developing human ventricle and adult atrium. Mutations in MYH7 are a common cause of hypertrophic cardiomyopathy, and end-stage hearts exhibited changes in MYH7/MYH6 ratio [29], which may be affected by DNMT3A. DNMT3A knockout also resulted in the accumulation of lipid vacuoles in the cardiomyocyte due to the aberrant activation of the glucose and lipid metabolism regulator peroxisome proliferator-activated receptor gamma (PPARγ), impaired glucose metabolism, and lower glycolytic enzyme expression, thus increasing the susceptibility to metabolic stress and precipitating heart failure [28].

In the transverse aortic constriction (TAC) model of pressure overload and heart failure, studies found that most DNA methylation and chromatin accessibility changes were observed 3 days after TAC, before heart failure develops, and localizes preferentially to intergenic and intronic regions [7]. Using the ChIP-qPCR technique, it was demonstrated that there was increased enhancer and promotor occupancy of GATA-binding protein 4 (GATA4) and NKX2.5 at the genes Itga9 and natriuretic peptide A (Nppa), respectively, increasing the transcription of these genes [7]. Nppa encodes the atrial natriuretic peptide (ANP), which is increased in heart failure to counter adverse cardiovascular actions of aldosterone and improve microcirculatory perfusion with cardioprotective effects [30].

DNMT1 is increased in mouse models with elevated homocysteine levels and may cause cardiac remodelling through DNA methylation [31]. The tissue inhibitor of matrix metalloproteinase 4 (TIMP4) is a modulator of matrix metalloprotease 9 (MMP9), which regulates the pathological cardiac remodelling process, and it has been demonstrated that TIMP4 is reduced in heart failure. In a heart failure model induced by the creation of an aorta-vena cava fistula in mice, it was shown that seven CpG islands in the TIMP4 promotor were methylated during the progression of heart failure, leading to epigenetic silencing and the resultant upregulation of MMP9, leading to cardiac remodelling [8]. Accordingly, in human patients with MI, raised MMP9 expression and the associated cardiac remodelling are associated with an increased risk of developing heart failure and complications, resulting in poor prognosis [32,33].

DNA methyltransferases such as DNMT3 and DNMT1 may mediate the cardiac remodelling in the development of heart failure through de novo methylation of DNA of non-coding regions, with impacts on the genetic expression of proteins with significant roles, such as MYH7, MYH6 and MMP9. Therefore, DNMT may be potential therapeutic targets that require further study [34].

## 4. Histone Modification in Heart Failure

Histones are basic proteins around which DNA wraps and are grouped into octamers typically consisting of two copies each of H2A, H2B, H3 and H4 subunits. Histones can undergo post-translational modifications in the amino-terminal tails, including methylation, acetylation, ubiquitination, sumoylation and phosphorylation, which regulate the degree of chromatin condensation, transcription factor binding and transcription elongation. Histone methylation on lysine and arginine residues in H3 and H4 are perpetuated by histone methyltransferases (HMTs), and the methyl groups are removed by demethylases in a dynamic process. Methyl groups are added to lysine or arginine residues on histones, which can lead to activation or repression of transcription depending on the degree and site of methylation. Histone acetylation also occurs on lysine residues and is controlled by histone acetyltransferases (HATs) and histone deacetylases (HDACs). Lysine acetylation typically results in transcription activation by relaxing the nucleosomal structure and increasing the accessibility to DNA-binding elements. HDACs are classified into four major classes, where Class I (HDAC 1, 2, 3 and 8) are widely expressed while Class II (HDAC 4, 5, 6, 7, 9 and 10) are cell type-restricted. The BET family of bromodomain (BRD) proteins read the acetylation pattern of histones and facilitate the binding of protein complexes that affect gene regulation. All four histone tails can be phosphorylated by protein kinases and dephosphorylated by phosphatases, and histone phosphorylation has a role in DNA damage repair, transcription regulation and chromatin remodelling [35]. Histone ubiquitination, where the 76 amino acid protein ubiquitin is post-translationally attached to lysine residues on the histone core proteins, is thought to lead to transcriptional activation [36]. Sumoylation involves the linking of small ubiquitin-like modifier (SUMO) moieties to the lysine residues on histones, which suppresses gene transcription and may be involved in DNA damage repair and chromatin remodelling [37].

### 4.1. Histone Methylation

Trimethylation of the histone H3 on lysine-4 (K4) and possibly lysine-9 (K9) is associated with heart failure in rat disease models. A genome-wide study using high-throughput pyrosequencing with ChIP products for H3K4 from human left ventricular tissue revealed differential H3K4 marking during the development of heart failure between functional and disabled cardiomyocytes, while the H3K9 mark profile was less dependent on disease status [38]. In end-stage non-ischemic dilated cardiomyopathy, the left ventricle expressed less H3K4 and H3K9 trimethylation compared to normal human left ventricle samples [39]. This was reversed after LVAD support, where H3K9 metheyltransferase is upregulated while demethylase is downregulated. ANP and brain natriuretic peptide expression also negatively correlated with H3K9 dimethylation and trimethylation, suggesting that histone methylation may have a role in end-stage DCM and reverse modelling. Jumonji domain-containing 1C (JMJD1C) is a histone demethylase that increases in expression during cardiac hypertrophy, leading to a decrease in H3K9 demethylation in humans and mice [9]. The inhibition of JMJD1C conversely reduces cardiac hypertrophy and fibrosis induced by angiotensin II and decreases TIMP1 transcription with pro-fibrotic activity [40]. Chaetocin is an H3K9 methyltransferase inhibitor that has been shown to improve the survival of Dahl sensitive rats, an animal model of heart failure, and reduce heart failure progression [41]. Histone methylation particularly at H3K4 and H3K9 is associated with heart failure and may have a role in reverse modelling. However, histone methylation as a therapeutic target requires further validation in humans and the safety inhibition of histone methylation is unclear.

### 4.2. Histone Acetylation

Histone acetylation is an important focus of heart failure epigenetics research and has been linked to cardiac hypertrophy. The overexpression of CREB-binding protein (CBP) and p300, which are HATs, leads to left ventricular myocyte hypertrophy, dilation and dysfunction via the acetylation of GATA4, which then increases the expression of hypertrophy-responsive genes such as Nppa and MYH7. The inhibition of p300 by curcumin reduced H3 and H4 acetylation and GATA4 DNA binding activity in a phenylephrine-induced hypertrophy rat model and prevented the deterioration of systolic function and HF in a salt-sensitive Dahl rat model [10]. Conversely, Class II HDACs inhibit the function of HATs and reduce cardiac hypertrophy. In mouse models, the Class II HDACs HDAC9 and HDAC5 suppressed cardiac hypertrophy [11], and knockout mice lacking HDAC5 and HDAC9 showed spontaneous cardiac hypertrophy with age and in response to constitutive calcineurin activation or pressure overload due to aortic constriction [12]. Mechanistically, this is via the inhibition of Mef2c by HDAC5/9, which promotes gene expression of pro-hypertrophy genes. HDAC5 and HDAC9 levels are regulated via phosphorylation by the stress-induced kinases calcium/calmodulin-dependent protein kinase (CaMK) and protein kinase D (PKD), and phosphorylated HDAC is transported out of the nucleus, thus allowing Mef2c to promote transcription through the binding of nuclear factor of activated T cells (NFAT) or GATA4 in cardiac stress. The N-terminal proteolytically derived fragment of HDAC, HDAC4-NT, was observed to be lower in the hearts of mice with heart failure, and its expression was increased with exercise [13]. Cardiomyocyte-specific deletion of the Hdac4 gene led to reduced exercise capacity, with increased cardiac fatigue, increased expression of Nr4a1 and activation of hexosamine biosynthetic pathway, affecting calcium sensor STIM1 and calcium handling. Increasing HDAC4-NT via virus-mediated transfer protected the mice myocardium from remodelling and failure. In mice lacking HDAC6, a Class IIb HDAC, myofibril stiffness was increased, while HDAC6 overexpression or ex vivo treatment with recombinant HDAC6 reduced myofibril stiffness in mouse, rat and human myofibrils [14]. HDAC6 may alter the titin compliance and thus may be another therapeutic target to modulate myocardial stiffness in pathological remodelling.

HDAC2, a Class I HDAC, may have the opposite effect by promoting genes associated with cardiac hypertrophy. HDAC Class I downregulates anti-hypertrophy genes including Kruppel-like factor 4 (Klf4) and inositol-5 phosphatase f (Inpp5f), promoting cardiac hypertrophy. HDAC2-deficient mice were resistant to pro-hypertrophy stimulation and showed resistance to the re-expression of fetal genes [15]. This may be due to the increased expression of the Inpp5f gene and the activation of glycogen synthase kinase 3beta (Gsk3b). The chemical inhibition of Gsk3b in HDAC-2 deficient mice rendered them to become sensitive to hypertrophic stimulation; thus, HDAC2 and Gsk3b may be potential targets in heart failure. HDAC1, another Class I HDAC, is also shown to increase cardiac fibrosis. In cardiac fibroblasts, the overexpression of peptidase inhibitor 16 (PI16) decreases nuclear HDAC1 after angiotensin II treatment, which leads to the increased acetylation of H3K18 and H3K27, and the inhibition of cardiac fibroblast proliferation and fibrosis-associated proteins [16]. Accordingly, transgenic mice with an overexpression of PI16 had a lower left ventricular mass when compared to wild-type mice in cardiac stress.

Trichostatin A is an inhibitor of Class I and II HDACs, and the injection of Trichostatin A in mice subjected to thoracic aortic banding to induce pressure overload resulted in a dose-responsive suppression of ventricular growth, with no evidence for cell death or apoptosis. There was also a reduction in fibrotic changes and collagen synthesis, and the preservation of systolic function with blunted hypertrophic growth, as shown by echocardiography and invasive pressure measurements [42]. Treatment with other HDAC inhibitors, such as API-D, a Class I HDAC inhibitor, also reduced cardiac hypertrophy and transition to heart failure in mice with thoracic aorta constriction, with significantly improved echocardiographic and hemodynamic parameters [43].

Class III HDACs, also known as sirtuins (SIRT), are generally protective against cardiac hypertrophy. The activity of sirtuins is dependent on the level of nicotinamide adenine dinucleotide (NAD^+^), and they interact with nuclear and mitochondrial proteins in metabolism and ATP synthesis. Sirtuins decrease oxidative damage, improve mitochondrial function and have a favourable effect against cardiac ageing. In response to pressure overload in mice, a low and moderate upregulation of Sirt1 (2.5- to 7.5-fold increases, respectively) appeared to attenuate cardiac hypertrophy, apoptosis and fibrosis, cardiac dysfunction and the expression of senescence markers [17]. However, high levels of overexpression (12.5-fold) instead promoted apoptosis and hypertrophy [17]. Sirt2 and Sirt6 may also prevent cardiac hypertrophy. The loss of Sirt6 in knockout mice led to increased H3K9 acetylation, allowing stress-responsive transcription factor c-Jun to interact more easily, activating the IGF-AKT signalling pathway, which resulted in cardiac hypertrophy [18]. In cardiac tissues from patients with heart failure and diabetes, endothelial Sirt6 expression is observed to be attenuated [19]. The restoration of Sirt6 expression in diabetic mice reduced diastolic dysfunction with reduced cardiac lipid accumulation [19]. This might be via the Sirt6-mediated deacetylation of H3K9 around the PPARγ promotor region, which reduces PPARγ expression and thus endothelial fatty acid uptake. The pharmacological activation of Sirt6 by MDL-800 reduced cardiac lipid accumulation and diastolic dysfunction in diabetic mice.

Histone acetylation and deacetylation are dynamic processes and play important roles in the early development and progression of heart failure. For example, in Dahl salt-sensitive rats where heart failure was induced by a high salt diet, the acetylation of H3K9 on the promotor of hypertrophic response genes was significantly increased in the hypertrophy stage. Conversely, the acetylation of H3K122 was not increased in the hypertrophy stage but was only evident in the heart failure stage [20]. This differential acetylation pattern led to an increase in interaction between chromatin remodelling factor BRG1 and p300 in the heart failure stage but not in the left ventricular hypertrophy stages.

Class II HDACs suppress cardiac hypertrophy while Class I HDACs and Class III HDACs are generally cardioprotective with the opposite effect. The dynamic changes in histone acetylation increase the challenges faced when investigating the role of histone acetylation and deacetylation in the development and progression and heart failure, and the timing of administration is an important consideration in the development of future epigenetic therapeutics.

### 4.3. BET-Family Bromodomain Proteins

The BET family of BRDs associates with acetylated chromatin to facilitate transcription activation, leading to downstream effects in cardiac hypertrophy, fibrosis and heart failure. Specifically, BRD4 has been shown to facilitate the expression of proinflammatory cytokines, proatherosclerotic targets, with roles in calcification, thrombosis and lipid metabolism. In heart failure, BETs may drive pathologic cardiac remodelling in cardiac stress. The inhibition of BRD4 by JQ1 in TAC models and phenylephrine-induced hypertrophy suppressed cardiomyocyte hypertrophy in vitro and cardiac remodelling in vivo [21]. The BETonMACE study is a phase 3 trial that investigated apabetalone, a BET inhibitor, in patients with acute coronary syndrome and diabetes mellitus. The primary outcome for this trial, namely cardiovascular death, MI or stroke was negative for apabetalone compared to the placebo [44]. However, a prespecified analysis found that apabetalone treatment was associated with fewer first and total hospitalizations for heart failure, and a lower combined rate of cardiovascular death and hospitalization for heart failure compared to the placebo [45].

Therefore, BRD4 may promote the expression of genes involved in cardiac hypertrophy and pathological remodelling, and BET proteins are potential therapeutic targets in heart failure, but this requires further study.

### 4.4. Histone Phosphorylation, Ubiquitination and Sumoylation

Studies on animal models and human heart tissues suggest that histone phosphorylation may be involved in the development of cardiac hypertrophy and fibrosis. Calcium/calmodulin-dependent protein kinase IIδ (CaMKIIδ) regulates the phosphorylation of H3 at serine-10 during pressure overload hypertrophy, and H3 serine-10 phosphorylation is increased in adult mouse ventricular myocytes and non-cardiac cells in early phases of cardiac hypertrophy. In CaMKIIδ deficient mice, the phosphorylation of H3 serine-10 was abolished after pressure overload [22]. CaMKIIδ deficient mice also showed reduced kinase activity towards HDAC4, which regulates stress-responsive cardiac gene expression and is protective against heart failure [23]. The potential role of CaMKIIδ as a therapeutic target requires further investigation.

Histone ubiquitination mechanisms have recently gained traction whereby alterations in these processes are associated with the development of cardiac diseases. The genetic deletion of the components of the E3 ubiquitin-protein ligase BRE1A complex responsible for H2B monoubiquitination (H2Bub1) in experimental models altered cilia motility and resulted in defective heart looping during embryogenesis, thereby precipitating congenital heart diseases [46]. Loss-of-function mutations resulting in impaired H2Bub1 were found to be enriched in patients with congenital heart diseases such as tetralogy of Fallot (ToF) and the transposition of the great arteries (TGA) [46], both of which are associated with significantly elevated risks of heart failure later in life, at approximately 80% by age 50 [47]. Similarly, those with isolated congenital valvular diseases or left-to-right shunts have up to a 30% chance of developing heart failure, and whether the loss of H2B ubiquitination directly contributes to heart failure pathogenesis and the potential underlying mechanisms should be explored. On the other hand, the ubiquitination of H2A on lysine 119 by the polycomb repressive complex 1 (PRC1) complex is associated with gene silencing and has been shown to contribute to reperfusion injury after myocardial ischemia [24]. An experimental inhibition of the PRC1 complex enhanced the expression of heat shock protein 27 (Hsp27) [24], which has well-established cardioprotective effects against oxidative stress and myocardial infarction [48,49]. Mechanistically, Hsp27 promotes glycolysis during myocardial ischemia through the induction of the nuclear factor kappa B (NF-κB) pathway, reduces mitochondrial production of reactive oxygen species (ROS) and prevents ferroptosis in response to cardiac ischemia [24]. Whilst heart failure is a common complication following MI, whether H2A ubiquitination contributes to heart failure development post-ischaemic events needs to be investigated. Intriguingly, several recent studies have identified ferroptosis to play key roles during heart failure pathogenesis [50,51], potentially providing a mechanistic link whereby the H2A ubiquitination-mediated suppression of Hsp27 may facilitate heart failure through enhanced ferroptosis. The protective effects of reducing H2A ubiquitination, for example through suppression of the PRC1 complex, may represent a potential novel therapeutic strategy in preventing post-cardiac ischemia heart failure, though further studies in this avenue are required.

Unlike other types of histone modifications, the potential roles of histone sumoylation in heart failure remain largely unexplored. Sumoylation is a type of post-translational modification on proteins and has important roles across a broad range of physiological processes such as cellular division, metabolism, DNA repair and epigenetic control of gene expression [52]. In humans, the sumoylation of histones is known to occur on all major histone subunits including H1, H2A, H2B, H3, H4 and the H2A.X variant and is typically associated with the suppression of gene expression [37]. Derangements in these pathways have been associated with a broad range of diseases including infection, cancer, autoimmune diseases and cardiovascular diseases such as heart failure [53,54,55,56,57]. The genetic deletion of SUMO1 in experimental models increased the risk of developing atrial and ventricular septal defects [58], and the overexpression of SUMO1 through adenovirus-mediated transfection in the TAC heart failure model prevented cardiac hypertrophy and protected against heart failure development [25]. On the other hand, SUMO2 and SUMO3 expression were found to be elevated in human patients with heart failure, and transgenic overexpression of SUMO2 in experimental models resulted in cardiomyopathy-associated heart failure [26]. However, the sumoylation status of histones has not been specifically investigated in the above studies.

Histone phosphorylation, ubiquitination and sumoylation are generally understudied and their potential roles in heart failure pathogenesis remain to be confirmed. CaMKIIδ, H2B ubiquitination and SUMO1 may reduce cardiac hypertrophic response, while H2A ubiquitination may contribute to cardiac injury. However, these studies were largely performed in mouse models and require further validation, although they may serve as a promising novel therapeutic strategy.

## 5. Future Directions

The dynamic and reversible nature of epigenetic changes could both be a therapeutic opportunity and a challenge. The change in DNA methylation in cardiomyocytes may orchestrate the modulation of gene expression in various stages of development, as shown by the study by Gilsbach et al. on newborn, adult and diseased hearts [59]. Histone modifications also interact with other epigenetic changes, such as DNA methylation to form an epigenetic pattern that leads to cardiac remodelling in pathological conditions, which are transient and context-dependent. For example, cardiomyocyte genes were sequentially demethylated from embryonic stages to newborn and adult stages, and demethylated genes that were transiently expressed in embryonic development were repressed by EZH2-mediated H3K27 trimethylation or de novo methylation mediated by DNMT3A/B. It was observed that heart failure-associated differentially methylated regions overlapped with enhancer or promoter regions marked by H3K27 acetylation, H3K4 methylation and H3K4 trimethylation in adult cardiomyocytes [59]. The methylation levels partially resembled the newborn CpG methylation pattern; were adjacent to genes involved in cardiac muscle cell development, cardiac morphogenesis and energy metabolism; and may be potential therapeutic targets to disrupt the development of the heart failure phenotype. In TAC-stressed mice, it was shown that Brg1 (nucleosome-modelling factor) is activated and recruits G9a/G1p (histone methyltransferase) and DNMT3, causing H3K9 and CpG methylation on MYH6 [60]. This leads to the silencing of MYH6 and impaired cardiac contraction, which is reversed by disruption of the Brg1, G9a or DNMT3. The overall epigenetic changes in heart failure from the epigenome are attractive novel targets for the treatment of heart failure (Table 2).

The most-studied targets are the HDACs, for example TSA, scriptaid and Givinostat, which inhibit Class I and II HDACs and reduce cardiac hypertrophy, fibrosis and arrhythmia in heart failure mice models [42,64,66]. HDAC inhibitors in human clinical trials are mostly limited in oncology, where severe side effects such as hematological toxicities have been reported [71]. Therefore, in heart failure, more cardiac selective HDAC inhibition may be required. Clinically used medications may also be repurposed; for example, valproic acid has been shown to reduce cardiac hypertrophy in mice models and inhibit Class I HDACs [67,68,69]. Other small molecules that target DNA methylation [34,61,62,63] such as HATs [10], BET proteins [21], histone methyltransferases [41] and histone demethylases [70] also show promise in preclinical studies. A single-cell epigenomic analysis of hearts exposed to BET inhibitors showed that cardiac fibroblasts, which are activated in the tissue stress response, showed differential toggling between the activated and quiescent states directly correlating with BET inhibitor exposure. MEOX1, a transcription factor downstream of BET proteins, was identified as a central regulator of fibroblast activation, and its inhibition may be a potential therapeutic target in fibrotic disease and cardiac dysfunction [72].

Epigenetic markers may act as biomarkers for cardiovascular disease, including heart failure. In a multi-omics study of myocardial tissue and blood from patients with dilated cardiomyopathy, 59 epigenetic loci were identified to be associated with dilated cardiomyopathy, 27 of which were replicated in an independent cohort [73]. The B9D1 gene hypomethylation was found in the heart tissue and blood of dilated cardiomyopathy, and mutations in B9D1 result in disturbed heart development from disrupted cliogenesis. The study by Meder et al. found that the B9D1 DNA methylation outperformed N-terminal-proBNP as a gold-standard marker for dilated cardiomyopathy in the study cohort [73]. However, studies were often limited by the small sample size and a lack of large multicentre studies to demonstrate clinical utility. Each epigenetic mechanism may play multiple roles in different cardiovascular diseases, while each disease involves complex interactions of epigenetic modifications, increasing the difficulty of identifying reliable biomarkers and therapeutic targets. Future advancements in technology and understanding of epigenetics are needed to feasibly introduce epigenetics biomarkers into clinical practice.

The development of imaging techniques to identify epigenetic signatures may further propel research in this field. [^11^C]Martinostat is a selective radioligand for Class I HDACs used in positron emission tomography (PET), which can detect HDAC expression in vivo. Preliminary studies showed that HDAC intensity measured by [^11^C]Martinostat uptake is higher in human myocardium than skeletal muscle or adipose tissue [74]. Further studies are planned to investigate the differential expression of HDAC in diseased hearts, for example in patients with aortic stenosis (NCT03549559). This may have an impact on the selection of patients for HDAC therapeutics based on HDAC expression, which may set the stage for future research into personalized epigenetic therapies in heart failure.

However, the transient nature of DNA and histone modifications may pose additional challenges in identifying the optimal timing and potential unintended consequences to epigenetic therapeutics. Epigenetic modifications may also have extensive downstream effects that are not fully understood, thus limiting our ability to predict the possible adverse effects. For example, chaetosin, a histone methyltransferase inhibitor, was found to prolong survival and restore mitochondrial dysfunction in Dahl salt-sensitive rats through the reversal of H3K9me3 elevation in thousands of repetitive elements, such as mitochondria-related gene PPARγ coactivator 1 alpha [41]. Chaetocin also has pleiotropic effects, including inducing oxidative stress, autophagy, mitochondrial membrane depolarisation and apoptosis, and is a potential drug candidate for cancer [75]. The impact of these effects in the context of heart failure and their long-term consequences is uncertain, significantly limiting the development of epigenetic therapies. Our current understanding of the complex epigenetic interactions remains in its infancy, and more research into the role of different epigenetic modifiers is needed to elucidate the potential benefits and toxicity associated with these therapies.

## 6. Conclusions

Epigenetics is a growing field with great promise as a novel therapeutic target in the treatment of heart failure, which remains a condition with a poor prognosis despite improvements in medical therapy. DNA methylation and histone modifications by methylation, acetylation, phosphorylation, ubiquitination and sumoylation are critical processes that establish an epigenetic pattern, which translates environmental stress into genetic expression, thereby affecting heart failure. They may be involved in the early processes of heart remodelling in response to cardiac stress, the development of heart failure in pathological remodelling and its progression. The dynamic changes in the epigenome in the disease process and the impact of these changes on the development of therapeutics require further elucidation. At present, clinical trials on epigenetic therapies for heart failure are limited, leaving much to be discovered about the potential efficacy and safety of altering the epigenome.

## Figures and Tables

**Figure 1 ijms-25-12010-f001:**
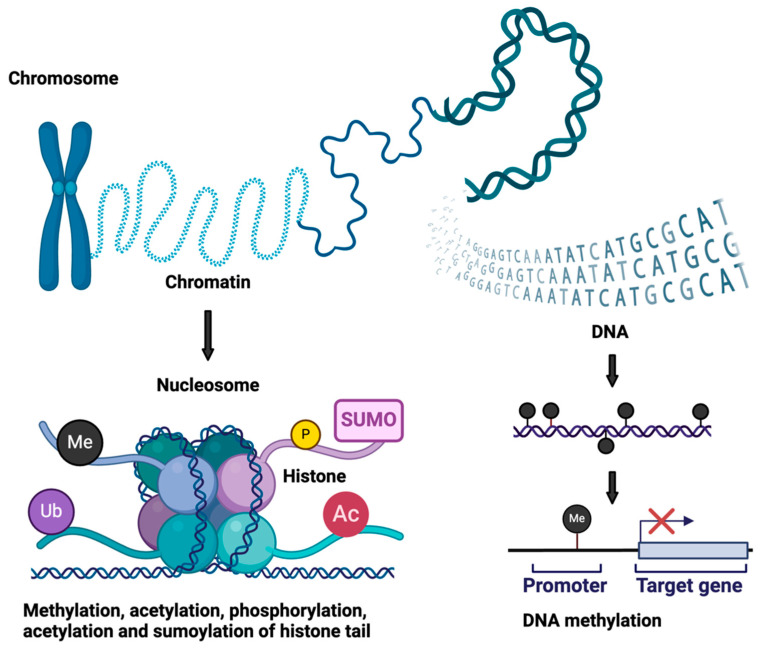
Summary of epigenetic modifications.

**Figure 2 ijms-25-12010-f002:**
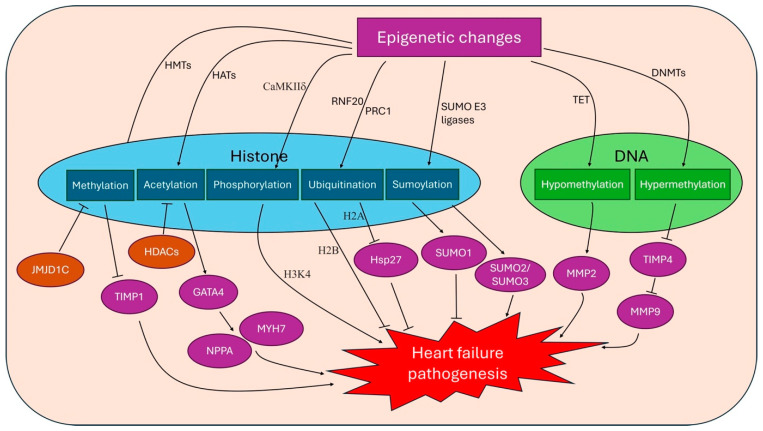
Overview of epigenetics changes in heart failure pathogenesis. HMTs, histone methyltransferases; HATs, histone acetyltransferases; CaMKIIδ, calcium/calmodulin-dependent protein kinase IIδ; RNF20, Ring Finger Protein 20; PRC1, polycomb repressive complex 1; TET, ten-eleven translocation; DNMTs, de novo methyltransferases; JMJD1C, Jumonji domain-containing 1C; TIMP, tissue inhibitor of matrix metalloproteinase; HDAC, histone deacetylases; GATA4, GATA-binding protein 4; NPPA, natriuretic peptide A; Hsp27, heat shock protein 27; MMP, matrix metalloprotease.

**Table 1 ijms-25-12010-t001:** Overview of the types of epigenetic modifications and proposed effects in heart failure.

Type of Epigenetic Modifications	Examples	Proposed Effects on Heart Failure	References
DNA hypomethylation	Hypomethylation of MMP2, CTGF increases their expression	Increased collagen turnover and fibrosis in heart failure	[6]
	Hypomethylation of Itga9 and natriuretic peptide A	Cardiac hypertrophy	[7]
DNA hypermethylation	Hypermethylation of TIMP4 leads to upregulation of MMP9, a regulator of cardiac remodelling process	Increased cardiac remodelling and heart failure	[8]
Histone methylation	Reduced H3K4 and H3K9 trimethylation, possibly linked to increased JMJD1C (a histone demethylase)	Non-ischemic dilated cardiomyopathy Increased cardiac hypertrophy and fibrosis induced by angiotensin II Increased atrial natriuretic peptic and brain natriuretic peptide expression	[9]
Histone acetylation	Acetylation of H3 and H4 by p300 (a HAT)	Phenylephrine-induced hypertrophy and deterioration of systolic function	[10]
	Class II HDAC inhibit HAT, e.g., HDAC5 and HDAC9	Reduce cardiac hypertrophy with age and in response to constitutive calcineurin activation or pressure overload	[11,12]
	HDAC4 deletion	Reduced heart failure, with reduced exercise capacity and increased cardiac fatigue	[13]
	Reduced HDAC6	Increased myofibril stiffness	[14]
	HDAC2 (Class I HDAC)	Promotes cardiac hypertrophy, downregulates anti-hypertrophy genes e.g., Klf4, Inpp5f, Gsk3b	[15]
	Decrease in nuclear HDAC1 (Class I HDAC) in cardiac fibroblast leads to increased acetylation of H3K18 and H3K27	Inhibition of cardiac fibroblast proliferation and fibrosis-associated proteins	[16]
	Class III HDAC or sirtuins e.g., Sirt1, Sirt2 and Sirt6 deacetylates H3K9	Decrease oxidative damage, improve mitochondrial function Reduced cardiac hypertrophy, apoptosis and fibrosis, cardiac dysfunction and expression of senescence markers in pressure overload Reduced cardiac lipid accumulation and diastolic dysfunction	[17,18,19]
	Acetylation of H3K9 on the promotor of hypertrophic response genes	Increased in heart failure induced by high salt diet, increased in hypertrophy stage	[20]
	BET-family bromodomain proteins interact with acetylated chromatin, e.g., BRD4	Drives pathologic cardiac remodelling in cardiac stress, inhibition of BRD4 suppressed cardiomyocyte hypertrophy	[21]
Histone phosphorylation	H3 serine-10 phosphorylation	Increased in early phases of cardiac hypertrophy with pressure overload	[22,23]
Histone ubiquitination	Ubiquitination of H2A on lysine 119 by PRC1 associated with gene silencing	Reperfusion injury after myocardial ischemia, uncertain effects in heart failure	[24]
Histone sumoylation	Overexpression of SUMO1	Prevent cardiac hypertrophy and heart failure development	[25]
	SUMO2 and SUMO3 overexpression	Cardiomyopathy associated with heart failure	[26]

BET, bromodomain and extra-terminal; BRD4, bromodomain-containing protein 4; CTGF, connective tissue growth factor; Gsk3b, glycogen synthase kinase 3beta; HAT, histone acetyltransferase; HDAC, histone deacetylases; Inpp5f, inositol-5 phosphatase f; JMJD1C, Jumonji domain-containing 1C; Klf4, Kruppel-like factor 4; MMP, matrix metalloprotease; PRC1, polycomb repressive complex 1; SUMO, small ubiquitin-like modifier; TIMP, tissue inhibitor of matrix metalloproteinase.

**Table 2 ijms-25-12010-t002:** Potential epigenetic therapeutics.

Compound	Epigenetic Target	Animal Model/Clinical Trial	Cardiac Effect	References
5-azcytidine	DNA methyltransferase inhibition	Hypertension-induced hypertrophy in rat	Reduced hypertrophy and fibrosis	[34]
5′-Aza-2′-deoxycytidine	DNA methyltransferase inhibition	Norepinephrine-induced hypertrophy in rat Spontaneously hypertensive rat	Reduced hypertrophy	[61,62]
RG108	DNA methyltransferase	TAC model in rat	Reduced hypertrophy	[63]
Curcumin	P300 histone acetyltransferase inhibitor	Phenylephrine-induced hypertrophy in rat Dahl salt-sensitive rat	Reduced hypertrophy and systolic dysfunction	[10]
TSA	Class I and II HDAC inhibitor	TAC model in mice Hop transgenic mice	Reduced hypertrophy and fibrosis	[42,64]
Scriptaid	Class I and II HDAC inhibitor	TAC model in mice	Reduced hypertrophy and fibrosis	[42]
Givinsostat (ITF2357)	Class I and II HDAC inhibitor	Uninephrectomy and DOCA-salt hypertensive mouse model, Dahl salt-sensitive rat	Reduced hypertrophy and fibrosis	[65,66]
Valproic acid	Class I HDAC inhibitor	TAC model in mice Hop transgenic mice Angiotensin II-induced hypertrophy in mice	Reduced hypertrophy	[67,68,69]
API-D	Class I HDACs	TAC model in mice	Reduced hypertrophy and fibrosis	[43]
Apabetalone	BRD4 inhibitor	Phase 3 clinical trial	Reduced heart failure hospitalizations	[45]
JQ1	BET proteins	TAC model in mice Phenylephrine-induced hypertrophy in mice	Reduced hypertrophy and fibrosis	[21]
JIB-04	Jumonji family (H3K9 demethylase) inhibitor	TAC model in mice	Reduced hypertrophy and fibrosis	[70]
Chaetocin	H3K9 methyltransferase inhibitor	Dahl salt-sensitive rat	Reduced hypertrophy	[41]

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
