# Peer review of "Epigenetics in Heart Failure"

_ijms, 2024, doi:10.3390/ijms252212010_

Round 1

Reviewer 1 Report

Comments and Suggestions for Authors

After reading the manuscript: “Epigenetics in Heart Failure” by Jamie Sin Ying Ho and all, a review of epigenetic changes involved in heart failure pathology, I have some comments:

1.    In a review that describe the epigenetic changes that influence the development of heart failure, a table that summarize the histone modification in heart failure should be very useful for the reader

  2. The paper can be read by cardiologists or doctors with limited knowledge of epigenetics. A brief description of epigenetic mechanisms may be useful to readers (for example, histone acetylation, phosphorylation, ubiquitination, sumoylation).

Author Response

Reviewer 1

  1. In a review that describe the epigenetic changes that influence the development of heart failure, a table that summarize the histone modification in heart failure should be very useful for the reader

We thank the reviewer for this suggestion, and we agree that this would enhance the clarity of the review. We have added a table to summarize the histone modifications in heart failure in Table 1.

Table 1: Overview of the types of epigenetic modifications and proposed effects in heart failure

Type of epigenetic modifications

Examples

Proposed effects in heart failure

Citation

DNA hypomethylation

Hypomethylation of MMP2, CTGF increases their expression

Increased collagen turnover and fibrosis in heart failure

(6)

Hypomethylation of Itga9 and natriuretic peptide A

Cardiac hypertrophy

(7)

DNA hypermethylation

Hypermethylation of TIMP4 leads to upregulation of MMP9, a regulator of cardiac remodelling process

Increased cardiac remodelling and heart failure

(8)

Histone methylation

Reduced H3K4 and H3K9 trimethylation, possibly linked to increased JMJD1C (a histone demethylase)

Non ischemic dilated cardiomyopathy

Increased cardiac hypertrophy and fibrosis induced by angiotensin II

Increased atrial natriuretic peptic and brain natriuretic peptide expression

(9)

Histone acetylation

Acetylation of H3 and H4 by p300 (a HAT)

Phenylephrine-induced hypertrophy and deterioration of systolic function

(10)

Class II HDAC inhibit HAT, e.g. HDAC5 and HDAC9

Reduce cardiac hypertrophy with age and in response to constitutive calcineurin activation or pressure overload

(11, 12)

HDAC4 deletion

Reduced in heart failure, with reduced exercise capacity and increased cardiac fatigue

(13)

Reduced HDAC6

Increased myofibril stiffness

(14)

HDAC2 (Class I HDAC)

Promotes cardiac hypertrophy, downregulates anti-hypertrophy genes e.g. Klf4, Inpp5f, Gsk3b

(15)

Decrease in nuclear HDAC1 (Class I HDAC) in cardiac fibroblast lead to increased acetylation of H3K18 and H3K27

Inhibition of cardiac fibroblast proliferation and fibrosis-associated proteins

(16)

Class III HDAC or sirtuins e.g. Sirt1, Sirt2 and Sirt6

Sirt6 deacetylates H3K9

Decrease oxidative damage, improve mitochondrial function

Reduced cardiac hypertrophy, apoptosis and fibrosis, cardiac dysfunction and expression of senescence markers in pressure overload

Reduced cardiac lipid accumulation and diastolic dysfunction

(17-19)

Acetylation of H3K9 on the promotor of hypertrophic response genes

Increased in heart failure induced by high salt diet, increased in hypertrophy stage

(20)

BET-family bromodomain proteins interact with acetylated chromatin, e.g. BRD4

Drives pathologic cardiac remodelling in cardiac stress, inhibition of BRD4 suppressed cardiomyocyte hypertrophy

(21)

Histone phosphorylation

H3 serine-10 phosphorylation

Increased in early phases of cardiac hypertrophy with pressure overload

(22, 23)

Histone ubiquitination

Ubiquitination of H2A on lysine 119 by PRC1 associated with gene silencing

Reperfusion injury after myocardial ischemia, uncertain effects in heart failure

(24)

Histone sumoylation

Overexpression of SUMO1

Prevent cardiac hypertrophy and heart failure development

(25)

SUMO2 and SUMO3 overexpression

Cardiomyopathy associated with heart failure

(26)

BET, bromodomain and extra-terminal; BRD4, bromodomain-containing protein 4; CTGF, connective tissue growth factor; Gsk3b, glycogen synthase kinase 3beta; HAT, histone acetyltransferase; HDAC, histone deacetylases; Inpp5f, inositol-5 phosphatase f; JMJD1C, Jumonji domain containing 1C; Klf4, Kruppel-like factor 4; MMP, matrix metalloprotease; PRC1, polycomb repressive complex 1; SUMO, small ubiquitin-like modifier; TIMP, tissue inhibitor of matrix metalloproteinase

  1. The paper can be read by cardiologists or doctors with limited knowledge of epigenetics. A brief description of epigenetic mechanisms may be useful to readers (for example, histone acetylation, phosphorylation, ubiquitination, sumoylation).

We thank the reviewer for this constructive comment. A brief description of DNA methylation is present in the “DNA methylation in heart failure” section and we have added a few sentences explaining rarer histone modifications such as ubiquitination and sumoylation in the “Histone modification in heart failure” section. A figure summarising the epigenetic modifications included in this review is added as Figure 1.

Histone ubiquitination, where the 76 amino acid protein ubiquitin is post-translationally attached to lysine residues on the histone core proteins, is thought to lead to transcriptional activation (14).  Sumoylation involves the linking of small ubiquitin-like modifier (SUMO) moiety to the lysine residues on histones, which suppresses gene transcription and may be involved in DNA damage repair and chromatin remodeling (15).” – page 10

Reviewer 2 Report

Comments and Suggestions for Authors

This review explores the role of epigenetic modifications in heart failure, including DNA methylation, histone modifications, and the potential for epigenetic therapies. It evaluates current evidence and therapeutic developments, emphasizing the need for further research to fully understand the dynamic role of epigenetics in heart disease progression. However, here are some concerns.

1) Although many pathways are mentioned, the mechanistic link between epigenetic changes and specific cardiac outcomes could be more thoroughly explained.

2) The dynamic and reversible nature of epigenetic changes, which could be both a therapeutic opportunity and a challenge, is not adequately addressed. More emphasis on how transient or context-dependent these modifications can be would help to portray a fuller picture of the complexity of epigenetic regulation in heart failure.

3) There is little discussion on the risks of manipulating the epigenome, such as unintended off-target effects or long-term consequences, which could be a significant limitation in the development of such therapies.

4) The potential for epigenetic changes to serve as biomarkers does not delve into the current limitations, challenges, or the level of validation that these biomarkers have achieved in the context of heart failure.

Comments on the Quality of English Language

There are minor grammatical errors and awkward phrasing in some sections, which could benefit from proofreading.

Author Response

Reviewer 2 

1) Although many pathways are mentioned, the mechanistic link between epigenetic changes and specific cardiac outcomes could be more thoroughly explained.

We thank the reviewer for this comment. In the current literature, our understanding of the mechanisms between epigenetic changes and observed cardiac outcomes is limited. Postulated mechanisms, where available, are added in the review and summarised in figure 2. In many studies, only associations in the epigenetic changes and change in phenotype of the heart failure animal models were observed, thus the underlying mechanisms were not elucidated. We have added a table (Table 1) to summarize the epigenetic changes and cardiac outcomes, to improve the clarity of the manuscript.

“The main de novo DNA methyltransferase in cardiomyocytes is DNMT3A which plays an important role in cardiac stress, with effects on glucose and lipid metabolism and cardiac contraction. DNMT3A knocked out in human cardiomyocytes resulted in impaired contraction kinetics associated with a higher atrial gene expression and lower MYH7/MYH6 ratio (8). MYH7 encodes β-myosin heavy chain (MHC), which is the major left ventricular MHC in the adult human, while MYH6 encodes α-MHC and is the major MHC in the developing human ventricle and adult atrium. Mutations in MYH7 are a common cause of hypertrophic cardiomyopathy, and end-stage hearts exhibited changes in MYH7/MYH6 ratio (9), which may be effected by DNMT3A. DNMT3A knock out also resulted in accumulation of lipid vacuoles in the cardiomyocyte due to aberrant activation of the glucose and lipid metabolism regulator peroxisome proliferator-activated receptor gamma (PPARγ), and impaired glucose metabolism and lower glycolytic enzyme expression, thus increasing the susceptibility to metabolic stress and precipitating heart failure (8).” – page 8 and 9

2) The dynamic and reversible nature of epigenetic changes, which could be both a therapeutic opportunity and a challenge, is not adequately addressed. More emphasis on how transient or context-dependent these modifications can be would help to portray a fuller picture of the complexity of epigenetic regulation in heart failure.

We thank the reviewer for this suggestion. We have further expanded our discussion on the dynamic and reversible nature of epigenetic changes particularly in the context of cardiac development and cardiac disease in the Future Direction section.

“The dynamic and reversible nature of epigenetic changes could both be a therapeutic opportunity and a challenge. The change in DNA methylation in cardiomyocytes may orchestrate the modulation of gene expression in various stages of development, as shown by the study by Gilsbach et al. on newborn, adult and diseased hearts (55). Histone modifications also interact with other epigenetic changes, such as DNA methylation to form an epigenetic pattern to lead to cardiac remodelling in pathological conditions, which are transient and context-dependent. For example, cardiomyocyte genes were sequentially demethylated from embryonic stages to newborn and adult stages, and demethylated genes that were transiently expressed in embryonic development were repressed by EZH2-mediated H3K27 trimethylation or de novo methylation mediated by DNMT3A/B. It was observed that heart failure-associated differentially-methylated regions overlapped with enhancer or promoter regions marked by H3K27 acetylation, H3K4 methylation and H3K4 trimethylation in adult cardiomyocytes (55). The methylation levels partially resembled the newborn CpG methylation pattern, and were adjacent to genes involved in cardiac muscle cell development, cardiac morphogenesis and energy metabolism, and may be potential therapeutic targets to disrupt the development of the heart failure phenotype. In TAC-stressed mice, it was shown that Brg1 (nucleosome-modelling factor) is activated and recruits G9a/G1p (histone methyltransferase) and DNMT3, causing H3K9 and CpG methylation on MYH6 (56). This leads to the silencing of MYH6 and impaired cardiac contraction, which is reversed by disruption of the Brg1, G9a or DNMT3. The overall epigenetic changes in heart failure form the epigenome, are attractive novel targets for treatment of heart failure. However, the transient nature of DNA and histone modifications may pose additional challenges in identifying the optimal timing and potential unintended consequences to epigenetic therapeutics. Our current understanding of the complex epigenetic interactions remains in its infancy, and more research into the role of different epigenetic modifiers are needed to elucidate the potential benefits and toxicity associated with these therapies.” – page 19

3) There is little discussion on the risks of manipulating the epigenome, such as unintended off-target effects or long-term consequences, which could be a significant limitation in the development of such therapies.

This is a valid point, and we have explored this at the end of the future directions section. We focused on the risks of manipulating the epigenome and unintended off-target effects or long-term consequences.

“However, the transient nature of DNA and histone modifications may pose additional challenges in identifying the optimal timing and potential unintended consequences to epigenetic therapeutics. Epigenetic modifications may also have extensive downstream effects that are not fully understood, thus limiting our ability to predict the possible adverse effects. For example, chaetosin, a histone methyltransferase inhibitor, was found to prolong survival and restore mitochondrial dysfunction in Dahl salt-sensitive rats, through the reversal of H3K9me3 elevation in thousands of repetitive elements, such as mitochondria-related gene PPARγ coactivator 1 alpha (20). Chaetocin also has pleiotropic effects including inducing oxidative stress, autophagy, mitochondrial membrane depolarisation and apoptosis, and is a potential drug candidate for cancer (71). The impact of these effects in the heart failure setting as well as long term consequences is uncertain, significantly limiting the development of epigenetic therapies. Our current understanding of the complex epigenetic interactions remains in its infancy, and more research into the role of different epigenetic modifiers are needed to elucidate the potential benefits and toxicity associated with these therapies.” – page 22

4) The potential for epigenetic changes to serve as biomarkers does not delve into the current limitations, challenges, or the level of validation that these biomarkers have achieved in the context of heart failure.

We thank the reviewer for raising the point about epigenetic changes as biomarkers in heart failure. Although several biomarkers involved in DNA methylation and histone modifications are identified, the evidence for these is limited to small clinical studies or preclinical animal studies. There is a lack of evidence in the multicentre clinical setting, thus limiting the ability to comment on their validation for the prediction of heart failure. We have added a paragraph discussing this in the Future Directions section.

“Epigenetic markers may act as biomarkers for cardiovascular disease, including heart failure. In a multi-omics study of myocardial tissue and blood from patients with dilated cardiomyopathy, 59 epigenetic loci were identified to be associated with dilated cardiomyopathy, 27 of which were replicated in an independent cohort (70). The B9D1 gene hypomethylation was found in the heart tissue and blood of dilated cardiomyopathy, and mutations in B9D1 result in disturbed heart development from disrupted cliogenesis. The study by Meder et al. found that the B9D1 DNA methylation outperformed N-terminal-proBNP as a gold-standard marker for dilated cardiomyopathy in the study cohort (70). However, studies were often limited by the small sample size and lack of large multicentre studies to demonstrate clinical utility. Each epigenetic mechanism may play multiple roles in different cardiovascular diseases, while each disease involves complex interactions of epigenetic modifications, increasing the difficulty of identifying reliable biomarkers and therapeutic targets. Future advancements in technology and understanding of epigenetics are needed to feasibly introduce epigenetics biomarkers into clinical practice.” – page 21

Comments on the Quality of English Language

There are minor grammatical errors and awkward phrasing in some sections, which could benefit from proofreading.

We thank the reviewer for this comment and have proofread the manuscript for grammatical errors.

Reviewer 3 Report

Comments and Suggestions for Authors

Please ref. the attachment

Comments on the Quality of English Language

Overall good, but grammatical and spelling errors here and there need to be corrected. 

Author Response

Review 3

The write-up is well structured with details of different studies outlined in the DNA methylation and the Histone modification sections. A figure/visual for DNA methylation and Histone modification would help readers who aren’t as familiar with epigenetic modifications.

We greatly appreciate the reviewer’s suggestion, and have added a figure to summarize DNA and histone modifications (Figure 1).

Figure 1. Summary of epigenetic modifications

At the end of each section please include a summary of the key findings from the studies listed in that section and potential applications of these findings.

We thank the reviewer for this suggestion to improve the clarity of this review. We have added a short summary at the end of each sub-section.

“DNA methyltransferases such as DNMT3 and DNMT1 may mediate the cardiac remodelling in development of heart failure, through de novo methylation of DNA of non-coding regions with impacts on genetic expression of proteins with significant roles, such as MYH7, MYH6 and MMP 9. Therefore, DNMT may be potential therapeutic targets which requires further study (16).” – page 10

“Histone methylation particularly H3K4 and H3K9 is associated with heart failure and may have a role in reverse modelling. However, histone methylation as a therapeutic target requires further validation in humans and the safety inhibition of histone methylation is unclear.” – page 12

“Class II HDACs suppress cardiac hypertrophy while class I HDACs and  Class III HDACs are generally cardioprotective with the opposite effect. The dynamic changes in histone acetylation increases the challenges faced when investigating the role of histone acetylation and deacetylation in the development and progression and heart failure, and the timing of administration is an important consideration in the development of future epigenetic therapeutics.” – page 15

“Therefore, BRD4 may promote expression of genes involved in cardiac hypertrophy and pathological remodelling, and BET proteins are potential therapeutic targets in heart failure, but this requires further study.” – page 16

“Histone phosphorylation, ubiquitination and sumoylation are generally understudied and their potential roles in heart failure pathogenesis remain to be confirmed. CaMKIId,  H2B ubiquitination and SUMO1 may reduce cardiac hypertrophic response, while H2A ubiquitination may contribute to cardiac injury. However, these studies were largely performed in mouse models and require further validation, although they may serve as a promising novel therapeutic strategy.” – page 19

Please add in a section/discuss the limitations/challenges with epigenetic modifications and epigenetic-based therapeutics.

We thank the reviewer for this suggestion, the limitations and challenges of epigenetic modifications and therapeutics are discussed in more detail in the Further Directions section.

“The dynamic and reversible nature of epigenetic changes could both be a therapeutic opportunity and a challenge. The change in DNA methylation in cardiomyocytes may orchestrate the modulation of gene expression in various stages of development, as shown by the study by Gilsbach et al. on newborn, adult and diseased hearts (55). Histone modifications also interact with other epigenetic changes, such as DNA methylation to form an epigenetic pattern to lead to cardiac remodelling in pathological conditions, which are transient and context-dependent. For example, cardiomyocyte genes were sequentially demethylated from embryonic stages to newborn and adult stages, and demethylated genes that were transiently expressed in embryonic development were repressed by EZH2-mediated H3K27 trimethylation or de novo methylation mediated by DNMT3A/B. It was observed that heart failure-associated differentially-methylated regions overlapped with enhancer or promoter regions marked by H3K27 acetylation, H3K4 methylation and H3K4 trimethylation in adult cardiomyocytes (55). The methylation levels partially resembled the newborn CpG methylation pattern, and were adjacent to genes involved in cardiac muscle cell development, cardiac morphogenesis and energy metabolism, and may be potential therapeutic targets to disrupt the development of the heart failure phenotype. In TAC-stressed mice, it was shown that Brg1 (nucleosome-modelling factor) is activated and recruits G9a/G1p (histone methyltransferase) and DNMT3, causing H3K9 and CpG methylation on MYH6 (56). This leads to the silencing of MYH6 and impaired cardiac contraction, which is reversed by disruption of the Brg1, G9a or DNMT3. The overall epigenetic changes in heart failure form the epigenome, are attractive novel targets for the treatment of heart failure. However, the transient nature of DNA and histone modifications may pose additional challenges in identifying the optimal timing and potential unintended consequences to epigenetic therapeutics. Our current understanding of the complex epigenetic interactions remains in its infancy, and more research into the role of different epigenetic modifiers are needed to elucidate the potential benefits and toxicity associated with these therapies.” – page 19

“Epigenetic markers may act as biomarkers for cardiovascular disease, including heart failure. In a multi-omics study of myocardial tissue and blood from patients with dilated cardiomyopathy, 59 epigenetic loci were identified to be associated with dilated cardiomyopathy, 27 of which were replicated in an independent cohort (70). The B9D1 gene hypomethylation was found in the heart tissue and blood of dilated cardiomyopathy, and mutations in B9D1 result in disturbed heart development from disrupted cliogenesis. The study by Meder et al. found that the B9D1 DNA methylation outperformed N-terminal-proBNP as a gold-standard marker for dilated cardiomyopathy in the study cohort (70). However, studies were often limited by the small sample size and lack of large multicentre studies to demonstrate clinical utility. Each epigenetic mechanism may play multiple roles in different cardiovascular diseases, while each disease involves complex interactions of epigenetic modifications, increasing the difficulty of identifying reliable biomarkers and therapeutic targets. Future advancements in technology and understanding of epigenetics are needed to feasibly introduce epigenetics biomarkers into clinical practice.” – page 21

“However, the transient nature of DNA and histone modifications may pose additional challenges in identifying the optimal timing and potential unintended consequences to epigenetic therapeutics. Epigenetic modifications may also have extensive downstream effects that are not fully understood, thus limiting our ability to predict the possible adverse effects. For example, chaetosin, a histone methyltransferase inhibitor, was found to prolong survival and restore mitochondrial dysfunction in Dahl salt-sensitive rats, through the reversal of H3K9me3 elevation in thousands of repetitive elements, such as mitochondria-related gene PPARγ coactivator 1 alpha (20). Chaetocin also has pleiotropic effects including inducing oxidative stress, autophagy, mitochondrial membrane depolarisation and apoptosis, and is a potential drug candidate for cancer (71). The impact of these effects in the heart failure setting as well as long term consequences is uncertain, significantly limiting the development of epigenetic therapies. Our current understanding of the complex epigenetic interactions remains in its infancy, and more research into the role of different epigenetic modifiers are needed to elucidate the potential benefits and toxicity associated with these therapies.” – page 22

Please check grammar and spellings throughout writing. In some sentences ‘a’ and ‘the’ are missing. Example: Abstract - ‘Heart failure is *a* clinical syndrome…’

We thank the reviewer for this comment and have proofread the manuscript for grammatical errors.

Reviewer 4 Report

Comments and Suggestions for Authors

The DNA methylation section was missing citations and there is no explanation of reading enzyme functions.

Comments on the Quality of English Language

 On page 9 it should be angiotensin instead of angiotension

Author Response

Reviewer 4

The DNA methylation section was missing citations and there is no explanation of reading enzyme functions.

We thank the reviewer for the comment. We have added appropriate citations and additional explanation regarding reading enzymes in the DNA methylation in heart failure section.

“Research focused on the role of DNA methylation in the pathophysiology of heart failure have caught attention in recent years (Figures 1 and 2, Table 1). There are various types of genomic DNA methylation, and the most commonly known is the addition of methyl group to the 5’ position of cytosine (5mC) by de novo methyltransferases (DNMT), which occurs mainly in the cytosinephosphateguanine (CpG) context (5). Classically, DNA methylation promotes gene silencing by blocking transcription factors from binding to the DNA directly, or through allowing binding of methyl-binding proteins (MBPs) which cause gene repression through interaction with a co-repressor complex. Conversely, oxidation of 5mC by ten-eleven translocation (TET) family of proteins generate 5-hydroxymethylcytosine (5hmC), which is associated with transcription activation. DNA methylation enzymes are broadly divided into writing, erasing and reading enzymes. Examples of writing enzymes include DNMT1, which adds methyl groups to cytosine residues mimicking the original DNA methylation pattern before DNA replication with high fidelity, while DNMT3a and DNMT3b are de novo methyltransferases that add methyl groups to naked DNA and establish new methylation patterns. Eraser enzymes remove methyl groups, while reading enzymes such as MBPs, UHRF (ubiquitin-like, containing PHD and RING finger domain) proteins and zinc-finger domain proteins recognize methyl groups and affect transcription factor binding, which can lead to gene silencing or activation (5).” – page 5

On page 9 it should be angiotensin instead of angiotension

We thank the reviewer for highlighting this. The spelling has been amended.

Round 2

Reviewer 2 Report

Comments and Suggestions for Authors

Thank you for your revisions. The major concerns have been fully solved in the current version, and I have no further questions.

Comments on the Quality of English Language

The English language quality is acceptable.